# Rett Syndrome and Other Neurodevelopmental Disorders Share Common Changes in Gut Microbial Community: A Descriptive Review

**DOI:** 10.3390/ijms20174160

**Published:** 2019-08-26

**Authors:** Elisa Borghi, Aglaia Vignoli

**Affiliations:** 1Department of Health Sciences, Università degli Studi di Milano, 20142 Milan, Italy; 2Child Neuropsychiatry Unit, ASST Santi Paolo Carlo Hospital, 20142 Milan, Italy

**Keywords:** neurodevelopment, Rett syndrome, autism spectrum disorder, gut microbiota, *Clostridium*, *Sutterella*

## Abstract

In this narrative review, we summarize recent pieces of evidence of the role of microbiota alterations in Rett syndrome (RTT). Neurological problems are prominent features of the syndrome, but the pathogenic mechanisms modulating its severity are still poorly understood. Gut microbiota was recently demonstrated to be altered both in animal models and humans with different neurodevelopmental disorders and/or epilepsy. By investigating gut microbiota in RTT cohorts, a less rich microbial community was identified which was associated with alterations of fecal microbial short-chain fatty acids. These changes were positively correlated with severe clinical outcomes. Indeed, microbial metabolites can play a crucial role both locally and systemically, having dynamic effects on host metabolism and gene expression in many organs. Similar alterations were found in patients with autism and down syndrome as well, suggesting a potential common pathway of gut microbiota involvement in neurodevelopmental disorders.

## 1. Introduction

The commensal microbial community inhabiting the mammalian gastrointestinal tract, microbiota, has been demonstrated to exert important functions both prenatally and postnatally [1]. Several studies on germ-free mice established that, at the local level, the presence of pioneering microorganisms triggers the correct development and maturation of the gut-associated lymphoid tissue. Indeed, in the absence of microbiota, animals fail to complete Peyer patches and mesenteric lymph nodes maturation and display longer and thinner villi with a reduced vascular network [2]. This intuitive and intimate relationship between the gut and its microbiota during development also applies to distant districts, with the brain as the most counterintuitive and intriguing. Microbial-derived signals are crucial for the physiological brain development, promoting neurotrophins secretion [3], microglia maturation [4], blood-brain barrier functionality (by enhancing tight-junction expression) [5], and correct myelination [6]. Brain development is tremendously vulnerable to environmental factors, and events occurring in early phases might have long-term effects later in life [7]. Beside the described role in early infancy, the microbiota-gut-brain axis was recently recognized to be a highly complex and tightly regulated bidirectional network. From top to bottom, it encompasses the central nervous system (CNS), the sympathetic and parasympathetic branches of the autonomic nervous system, the enteric nervous system, and the neuroendocrine and neuroimmune systems [8]. This complex network is enriched by the gut microbial community exerting a bottom-up modulation by interacting mainly with the vagus nerve and the neuroendocrine and neuroimmune systems [9].

In this review, we will focus on gut microbiota changes in Rett syndrome (RTT), the most prevalent neurodevelopmental disorder in females, whose pathogenic mechanisms are still poorly understood and whose genetic background cannot fully explain the phenotypic heterogeneity.

## 2. Gut Microbiome Alterations in Neurodevelopmental Disorders

Neurodevelopmental disorders (NDs) are chronic disorders that affect CNS function during the developmental period in the domains of motor skills, cognition, communication, and/or behavior. The diagnosis of NDs is primarily clinical, based on a constellation of behaviors/symptoms that are specified in the diagnostic and statistical manual of mental disorders (DSM) [10]. The most common ND worldwide is autism spectrum disorder (ASD), for which several studies, conducted since 2000 in different geographical regions and by different teams, have estimated a median prevalence of 62/10,000 [11].

ASD is an extremely heterogeneous neurodevelopmental disorder and its onset can vary. Some children show signs from early infancy, while others exhibit symptoms of regression around 18–24 months. Other specific features may be recognizable only at an older age. Clinically, ASD is characterized by stereotypical behaviors and a deficit in social interaction and communication [12]. In addition to the neuropsychiatric features, children with ASD often experience gastrointestinal (GI) problems that are more frequent and more severe compared with children from the general population. Functional constipation is the most common issue [13], followed by diarrhea/chronic diarrhea, abdominal pain or discomfort, bloating, gas or flatulence, soiling, incontinence or bedwetting, reflux or heartburn, nausea or vomiting, and pain or difficulties in bowel movement [14]. Gastrointestinal symptoms strongly correlate with the severity of autism and with increased irritability, anxiety, and social withdrawal [15,16]. This evidence elicits researches focusing on the connection between ASD and the gut microbiota. Intriguingly, an altered composition of gut microbiome has been demonstrated in subjects with ASD [17,18,19,20], suggesting a possible link between dysbiosis (an aberrant shift in microbial ecology) and neurodevelopmental diseases. Despite differences in cohorts, study designs, and methodologies, a common microbial signature has been identified (reviewed by Ding et al. [21] and by the recent systematic review by Liu et al. [22]). ASD-related major changes involve an increase in the relative abundance of *Clostridium* spp. and *Sutterella* spp., *Bacteroides* spp. and *Desulfovibrio* spp. and a depletion in *Bifidobacterium* spp., *Blautia* spp., *Prevotella* spp., and *Veillonella* spp. [19,23,24]. The *Clostridium* hypothesis, suggesting the involvement of toxigenic bacteria in the onset of the ASD regressive form, was first suggested by Ellen Bolte [25] and later supported by other authors [26,27]. Indeed, clostridia are spore-forming and can exert a toxic activity by producing exotoxins, phenol, and p-cresol [27,28]. P-cresol has been postulated as a possible urinary biomarker in ASD and its concentration correlates with constipation (slow fecal transit) [29]. Other microbial-derived metabolites have been linked to ASD in recent years, including propionate. Propionate is a short-chain fatty acid (SFCA) and one of the main products of microbial fermentation, together with acetate and butyrate. SCFAs are utilized by the host as an energy source and exert other physiological functions. SCFAs participate in glucose homeostasis, affect lipid metabolism, regulate the immune system and inflammatory response, and serve as signaling molecules [30].

Intracerebroventricular injection of propionate in rodents was found to cause social impairment [31], as well as behavioral, neuropathological, and biochemical changes associated with ASD [32]. However, the contribution of microbial metabolites to ASD still must be elucidated, taking into account that fecal SCFAs were reported to be either lower or higher in ASD children [15,33,34]. 

Among genetic NDs, down syndrome (OMIM #190685, trisomy 21–DS) is the most frequent cause of intellectual disability (ID). ASD features and the high prevalence of gastrointestinal system alterations (about 50%) often characterize DS individuals [35]. Despite DS incidence and the suggestive features of a possible gut microbiota modulation, only one study has investigated the DS gut microbial community thus far. Biagi and coworkers [36] compared 17 DS adults with a matched group of healthy controls. The authors reported specific changes in the microbial community of individuals affected by DS, with some of them being associated with behavioral traits. Although DS gut microbiota does not significantly differ in the relative abundance of the dominant microbial families (i.e., *Ruminococcaceae, Lachnospiraceae, Clostridiales, Bifidobacteriaceae*, and *Bacteroidaceae*), some subdominant taxa are differently represented in DS compared with healthy controls. In particular, the relative abundance of *Parasporobacterium* spp. and *Sutterella* spp. is increased, whereas *Veillonellaceae* is reduced. Besides, when considering the severity of neurobehavioral manifestations in DS patients, *Sutterella* was positively correlated with the aberrant behavior checklist (ABC) total score, while no correlation was found between ABC and *Parasporobacterium* or *Veillonellaceae*.

## 3. Gut Microbiome Alterations in Rett Syndrome

Although Rett syndrome (RTT) is no longer categorized as a pervasive developmental disorder, patients share certain features with ASD. RTT (OMIM #312750) is an X-linked dominant neurodevelopmental disorder and one of the most common causes of ID in females. Of all cases, 90–95% are associated with pathogenic variants in the *MECP2* (Methyl CpG Binding Protein 2) gene, encoding a chromatin-associated protein that can both activate and repress transcription [37,38]. The RTT core phenotype mainly consists of neurological problems (partial/complete loss of acquired purposeful hand skills; partial/complete loss of acquired spoken language; gait abnormalities; stereotypic hand movements) [39]. Most girls affected by RTT experience epileptic seizures, with epilepsy representing a major concern of RTT caregivers and having a crucial impact on children and their family’s quality of life [40]. In addition to epilepsy, parental stress is affected by their daughters’ GI pain experience [41]. Indeed, one of the major issues impacting patients affected by RTT on a daily basis is represented by gastrointestinal and nutritional problems [42]. Families reported a series of different symptoms, such as straining with bowel movements, the passage of hard stools, constipation, and prolonged feeding time or chewing difficulty in more than 50% of RTT patients [42]. More recently, cholelithiasis, or gallbladder disease, has been reported to be relatively frequent in RTT, and should be considered one of the causes of abdominal pain [43]. Due to the high frequency of gastrointestinal comorbidities, recommendations for managing GI symptoms have been recently developed [44]. 

Despite the similar features with ASD and the observation of the frequent gastrointestinal discomfort reported by RTT caregivers, the study of possible microbial alterations characterizing RTT gut microbiota is still relatively new. Only two Italian groups have investigated this aspect in RTT girls thus far. The characteristics and main results of these studies are outlined in Table 1. 

Because of the higher vulnerability to respiratory infections related both to RTT itself and to the different associated comorbidities (e.g. scoliosis, epilepsy, ID) [45], subjects with RTT may be exposed to several antibiotics to prevent/cure recurrent respiratory infections. Antibiotics are responsible for altering the gut microbiome. For this reason, in the studies conducted on RTT girls, a common exclusion criterion was an antibiotic assumption in the three months preceding the collection of the sample.

RTT gut microbiota is characterized by a reduction in α-diversity [46,47]. α-diversity metrics allow studying the richness, i.e., the number of unique microbial taxa within a given sample, as well as the evenness, how uniformly the unique taxa are distributed, and the phylogenetic relatedness between them. The loss of microbial diversity is considered a hallmark of dysbiosis, paving the way to a reduced gut microbiota resilience [48]. Thus, the gut microbial ecosystem of RTT girls can be considered intrinsically more susceptible to disease status. The decrease in diversity observed in RTT patients is more pronounced in severe phenotypes, as patients with a lower diversity have higher clinical severity scale (CSS) scores [47]. 

β-diversity (Unifrac distances) analyses, which measure dissimilarities in microbial community composition between samples, showed that RTT gut microbial community clustered according to the disease status. On the one hand, Strati and coworkers grouped RTT patients according to GI dysfunction and observed that health status, not GI symptoms, distinguish the patients from healthy controls, whereas constipated RTT and non-constipated RTT were not statistically different. On the other hand, the severity of the disease further stratified RTT patients [47]. In particular, unweighted Unifrac metric, which considers both rare and common bacterial taxa, highlighted a significant dissimilarity between the RTT and healthy subjects, suggesting that, as described for DS [36], subdominant taxa are differently represented. The analyses on the relative microbial abundance at different taxonomic levels showed an enrichment in *Erysipelotrichacea*e in RTT patients and, at the genus level, of *Clostridium* spp., *Sutterella* spp., and *Escherichia* spp. [46,47]. Other taxa were found to be discordant between the two studies, i.e., *Bifidobacterium* and *Bacteroides*. Bifidobacteria data could suffer from an age-related abundance. Indeed, Strati et al. enrolled younger girls affected by RTT, with a consistent number of children younger than 10 years old. This taxon is well-known to inversely correlate with age, with the highest abundance during lactation [49]. Like the microbial alterations reported in ASD patients, Borghi et al. observed a decrease in *Bacteroides* spp. [47]. This genus, as well as Bacteroidetes, are reported to be negatively correlated with body-mass index (BMI) [50,51,52]. BMI values were lower in the RTT group, justifying this observation. Anthropometric measurements were not specified in the second cohort. Despite differences in the observed taxa relative abundances in the two studies, a common increase in the fecal concentration of branched-chain fatty acids (iso-butyrate and iso-valerate-BCFAs) and, to a lesser extent, of propionate and butyrate, has been observed. BCFAs are byproducts of protein degradation, especially of animal proteins that are rich in branched-chain amino acids, by proteolytic bacteria such as Bacteroidetes. Dietary survey [47] showed a higher protein consumption, mainly due to higher animal protein intake, and a lower fiber intake in RTT compared with healthy controls. No specific dietary recommendations for RTT patients have been reported so far, but proteins might be preferred for the increased ratio nutrients/volume, considering eating difficulties, palatability, and texture. Beside BCFAs, proteolysis results in the production of phenolic and indolic compounds. As already mentioned for ASD, p-cresol, the main phenolic product, is a toxic compound and has been demonstrated to be able to alter the intestinal barrier permeability [53]. Although no study has evaluated its fecal or urinary concentrations in RTT subjects to date, the enrichment in gut microbiota in the main producing species, i.e., *Clostridium* spp. [46,47] *Bacteroides* spp. [47], suggests a possible increase of p-cresol also in RTT.

Mouse models of RTT are available and they recapitulate several neurological features observed in patients [54], including altered cortical rhythms and enhanced susceptibility to epileptic seizures [55,56]. Moreover, severe modifications in the colon organization are present in MeCP2 mutant mice [57], suggesting that alterations occurring at the gut level, including gut microbiome, could be associated with the classical neurological impairments observed in RTT mice. Despite this evidence, the gut microbiota of RTT mouse models has not been explored to date, and the aspects related to the gut microbiome-brain axis in this murine model have not been examined.

## 4. Gut Microbiome Alterations in Epilepsy

Both RTT and ASD are often comorbid with seizure disorders [58]. Although it is not mentioned in the diagnostic criteria [39], up to 70% of girls affected by RTT experience frequent seizures, and 30% of them are resistant to the available antiepileptic drugs (AEDs) [59]. In classic RTT, the mean age of epilepsy onset is 4 years. Seizure frequency and severity usually have an age-dependent course, as girls aged 10–14 years are the most difficult to treat and often require AED polytherapy [60]. Recent findings from the Rett Natural History Study pointed out that seizures in RTT may show a pattern of remission and relapse within the lifetime. Specific *MECP2* mutations are not significantly associated with either seizure prevalence or seizure severity, although seizure prevalence is associated with disease severity [61].

The potential role of gut microbiota in epilepsy is currently emerging. He et al. [62] studied a patient with a 17-year history of epilepsy, which improved after fecal microbiota transplantation treatment for Crohn’s disease. More recently, the gut microbial community in patients with drug-resistant epilepsy was found to be significantly altered with an abnormally increased abundance of rare bacteria mainly belonging from the phylum Firmicutes. On the contrary, the gut microbiome of patients with drug-sensitive epilepsy was more similar to that of healthy controls. Therefore, it can be speculated that gut dysbiosis may be involved in the pathogenesis of drug-resistant epilepsy [63]. In particular, patients with seizures responsive to AEDs have a greater amount of bifidobacteria and lactobacilli, which could be interpreted as protective factors for epilepsy [63]. Indeed, a recent study on drug-resistant epilepsy showed that probiotic supplementation with a mixture of bifidobacteria and lactobacilli reduces both seizures and sCD14 serum concentration, a recognized marker for bacterial translocation [64].

Many drugs have been demonstrated to modulate or alter the gut microbial community. In particular, compounds targeting the nervous system exhibited a significant anticommensal activity on a broad range of microorganisms [65]. Although the direct role of AEDs has not been investigated to date, it is plausible, although not yet elucidated [47].

Other pieces of evidence based on diet modification support the possible role of gut microbiota in modulating epilepsy. The ketogenic diet (KD) is a high-fat, adequate-protein, low-carbohydrate diet used as a treatment for neurometabolic disorders, such as glucose transporter 1 (Glut1)-deficiency syndrome, and for drug-resistant epilepsy. Considering the unbalanced macronutrient composition of the KD, it is expected to induce some changes in gut microbiota. Tagliabue et al. [66] evaluated the gut microbiota composition in six children with Glut1-deficiency syndrome after three months of KD treatment and did not find statistically significant differences in Firmicutes and Bacteroidetes. However, they found a statistically significant increase in *Desulfovibrio* spp., a subdominant taxon reported in inflammatory bowel disease and other inflammatory conditions [67].

Xie and coworkers [68] found that the gut microbiota of Chinese epileptic children differed from age-matched healthy infants and reported that the KD strongly improved gut microbiota alterations, promoting seizure reductions. Another study, which enrolled 20 children with refractory epilepsy, observed distinctive microbial changes in KD-responder (seizure-free or ≥ 50% of seizure reduction) and non-responder patients (<50% of seizure reduction). The KD-responders were characterized by an increase in the relative abundance of Bacteroidetes, whereas non-responders showed a significant increase in the relative abundance of the Firmicutes (*Clostridiales*, *Clostridia, Ruminococcaceae, Lachnospiraceae, Alistipes*, and *Rikenellaceae*) [69]. A further study on KD treatment for severe epilepsy in Swedish children, which applied a shotgun metagenomic DNA sequencing approach, found that alpha diversity was not significantly changed by diet. The study highlighted a decrease in the relative abundance of Actinobacteria and Firmicutes, and an increase of Bacteroidetes and Proteobacteria [70]. Intriguingly, microbiota changes were linked to functional changes, particularly in relation to carbohydrates metabolism pathways. 

The KD has been investigated as a possible intervention in RTT [71], especially in the *CDKL5*-related variant, which is characterized by multiple seizure-type epilepsy and a poor response to AEDs [72]. Major concerns to a broad application of the KD are the reported poor long-term efficacy and lack of adherence. 

Despite the growing number of studies on the KD and its impact on microbiota in epilepsy, the findings should be interpreted with caution because of the small number of patients involved, the different epilepsy etiology, and the specific diet composition. Nevertheless, the accumulating evidence suggests the need for a better understanding of the state of dysbiosis to establish strategies to possibly counterbalance it. 

## 5. Discussion

Intriguingly, neurodevelopmental disorders sharing clinical features display a microbial signature or are at least enriched/depleted in the same taxa. The major overlaps, both clinical and microbial, are highlighted in the Venn diagram (Figure 1).

The diagram was generated (http://bioinfogp.cnb.csic.es/tools/venny/index.html) considering only microbial changes reported in most studies (i.e., discarding discordant findings).

In particular, an enrichment in *Sutterella* spp. has been reported in all the reviewed NDs. A similar trend was also observed in attention-deficit/hyperactivity disorder (ADHD) [73] and in rodent models with Alzheimer’s disease [74] and ASD [75]. *Sutterella* belongs to Beta-proteobacteria, gram-negative bacterium. Due to the reported higher relative abundance in several human diseases, Hiippala and coworkers [76] assessed its in vitro pro-inflammatory activity and compared it with the well-known *Escherichia coli* (gamma-Proteobacteria). *Sutterella* demonstrated only a mild pro-inflammatory activity on intestinal epithelial cells, which was not sufficient to induce GI homeostasis alterations. On the other hand, this taxon was demonstrated to efficiently adhere to enterocytes [77], and the effect of a direct cross-talk with these cells in the frame of the gut-brain-axis warrants further elucidation. 

Similarly, RTT and ASD subjects share an increase in the relative abundance of *Clostridium* spp. This taxon, together with *Bacteroides* spp., is well-known for its proteolytic ability, which results in potentially toxic compounds that can impact on gut homeostasis and permeability [53] and can affect the survival of other beneficial microorganisms, such as lactic bacteria [77]. Bifidobacteria and lactobacilli are known to be able to secrete neurotransmitters, particularly the gamma-aminobutyric acid (GABA) [78]. A decrease in GABA and its signaling is suggested to be involved in many ND-associated clinical features, i.e., stereotypies, hypersensitivities, and seizures [79,80,81]. Moreover, the same taxa have been shown to promote intestinal mucosa integrity [82], decreasing local and systemic inflammatory status. CNS is highly susceptible to inflammation, contributing to the pathogenesis of RTT [83] and to associated comorbidities. In particular, neuroinflammation has been demonstrated to trigger seizures in patients with epilepsy [84]. A direct role of gut microbiota in eliciting inflammation and, in turn, seizure occurrence, has been postulated and investigated in animal models [85]. 

Although speculative conclusions about microbiota-dependent modifications in brain function and behaviors have been raised over recent years (summarized in the context of NDs in Figure 2), no studies have directly assessed the molecular or neurophysiological mechanisms responsible for neurodevelopmental alterations characteristic of NDs [86].

Most of the literature has focused either on the clinical aspect or on the microbiome alterations. Nevertheless, NDs are characterized by multifaceted phenotypes that might contribute to the observed differences in the microbiota among various cohorts. A comprehensive approach, combining detailed clinical description with microbiological data, is needed to better understand underlying relationships between symptoms and specific microbial alterations.

## 6. Conclusions

Although microbiome studies in RTT patients are in their early stages, several authors have investigated this aspect in ASD, which shares several clinical similarities with RTT. Main changes in the gut microbial community suggest enrichment in pro-inflammatory species that could promote alterations in the gut permeability and its barrier function. Within the context of the gut-brain axis, these microbiota-dependent modifications could trigger common behavioral and neurodevelopmental features. 

The observation of the presence of a dysbiosis in patients with NDs, especially ASD, triggers the development of innovative therapeutic strategies throughout microbiome-based treatment. Nevertheless, clinical trials with probiotic supplementation are currently limited and lack standardized probiotic regimen, i.e., different administered strains or concentrations and duration of treatment. No clinical trials have been carried out on RTT cohorts thus far.

In addition to probiotic intervention, a careful diet survey on large cohorts of RTT patients would allow the development of diet recommendations that might mitigate microbiome alterations per se. 

## Figures and Tables

**Figure 1 ijms-20-04160-f001:**
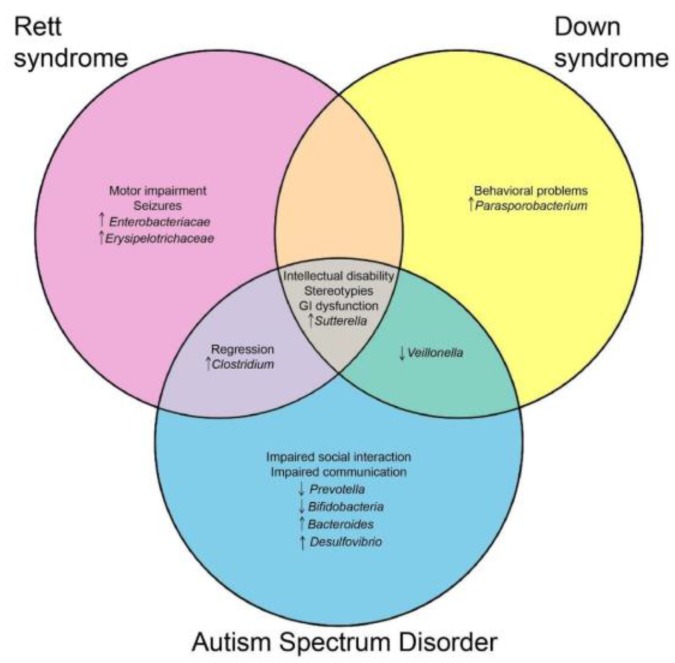
Venn diagram presenting the shared clinical and microbial features between Rett Syndrome, autism spectrum disorder, and down syndrome.

**Figure 2 ijms-20-04160-f002:**
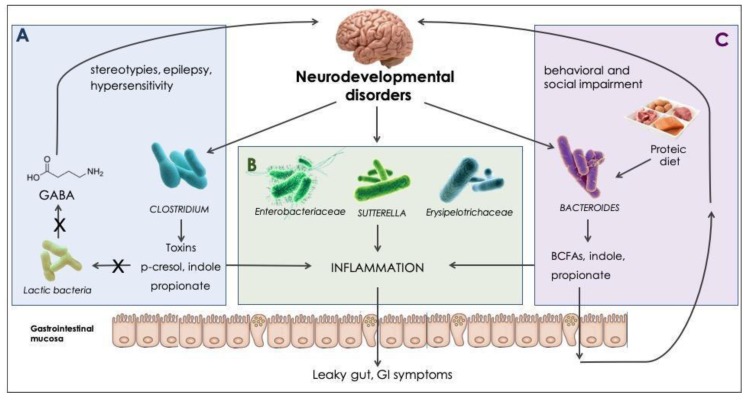
Microbial modulation in neurodevelopmental disorders. Effects driven directly or indirectly by gut microbes, found to be increased in NDs and their products. (**A**) Clostridia are spore-forming, toxigenic bacteria, with proteolytic ability [53]; digestion of aromatic amino acids results in phenol, p-cresol, and indole production. High indole concentrations can suppress bifidobacteria and lactobacilli growth [77]. Lactic bacteria are known to endogenously produce gamma-aminobutyric acid (GABA) [78]. GABA decrease has been reported as an underlying mechanism for stereotypies occurrence [79], ASD hypersensitivity [80], and epilepsy [81]. (**B**) *Enterobacteriaceae* [87], *Sutturella* spp. [76], and *Erysipelotrichaceae* [88] have been reported to exert a pro-inflammatory effect, which can alter gastrointestinal homeostasis and alter the gastrointestinal (GI) barrier permeability. (**C**) *Bacteroides* are nutritionally versatile bacteria, with both saccharolytic and proteolytic activity [53]; a rich-protein diet promotes the production of branched-chain fatty acids (BCFAs) and propionate [89]. High propionate concentrations have demonstrated to exert behavioral and social impairment in animal models [31].

**Table 1 ijms-20-04160-t001:** Characteristics of gut microbiota studies in Rett syndrome (RTT).

Reference	Cohort	Mean Age (years)	Age Range	GI^a^ Symptoms (%)	Methods	Results (RTT vs HC^b^)
Strati et al., 2017	50 RTT	12.6 ± 7.3	1.5–32	68%	16S rRNA sequencing: V3-V5 hypervariable regions; 454 pyrosequencing platform; Greengenes Version 13.05	↓ α-diversity^c^ β- diversity^d^ (*p* ≤ 0.003) ↑ Actinobacteria ↓ Bacteroidetes ↑ *Bifidobacterium* spp. ↑ *Clostridium* spp.*, Enterococcus,* *Erysipelotrichaceae incertae sedis, Escherichia/Shigella* and *Megasphaera* ↑ *Candida* spp.
	(MECp2; n = 48) (CDKL5; n=2)					
	29 HC^b^	17.4 ± 9.6	1.5–32			
					SCFA^e^ quantification: Gas chromatography	↑ Propionate, *iso*-butyrate and *iso-*valerate (in feces)
Borghi et al., 2017	8 RTT	23.0 ± 8.7	9–39	100%	16S rRNA sequencing: V3-V4 hypervariable regions; MiSeq Illumina platform; Greengenes Version 13.8	↓ α-diversity^c^ (a severity-related decrease) β- diversity^d^ (*p* ≤ 0.05, only severe RTT vs HC) ↑ *Bacteroides* spp. ↑ *Clostridium* spp. ↑ *Sutterella* spp. ↓ *Faecalibacterium* spp. ↓ *Roseburia* spp. ↓ *Prevotella* spp. ↑ *Enterobacteriacae, B**acteroidaceae* and *Erysipelotrichaceae* (severity-related)
	(MECp2; n = 8)					
	10 HC^b^	24.5 ± 6.6	18–41			
					SCFA^e^ quantification: Gas chromatography	↑ Propionate, butyrate, *iso*-butyrate and *iso-*valerate (in feces)

^a^GI, Gastrointestinal tract; ^b^HC, healthy controls; ^c^α-diversity, the diversity in the bacterial composition within each sample; ^d^β- diversity, the diversity between sample groups; ^e^SCFA, short-chain fatty acids.

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
