# Peer review of "Rett Syndrome and Other Neurodevelopmental Disorders Share Common Changes in Gut Microbial Community: A Descriptive Review"

_ijms, 2019, doi:10.3390/ijms20174160_

Round 1
Reviewer 1 Report
The central role of the brain-gut-microbiome axis (BGM) in neuronal development is well documented. A lot of recent studies have focused on the role of gut microbiome in regulating anxiety, depression, repetitive compulsive behavior etc. The purpose of this review is to explore the contribution of gut microbiome on RTT phenotype. The authors mostly review studies that analyze the composition of gut microbiome between healthy individuals and patients diagnosed with Autism Spectrum or RTT.
The authors start by providing a brief introduction of the importance of BGM axis and how the balance between the three components is important for neuronal development and function. The introduction section is well written and provides sufficient context for the rest of the review.
Table 1 in this article is very helpful and highlights the key points of the two studies that compare gut microbiome of RTT patients with that of healthy individuals. The authors analyze these two studies and point out that even though there are no significant differences in the microbiome composition on the whole, both the studies found that certain non-dominant taxa were enriched or depleted in RTT patients compared to healthy cohorts. The authors also discuss changes in the gut microbiome of Autism patients as well as patients suffering from epileptic seizures and how such studies can contribute to our understanding of the BGM axis in RTT.
Figure 1 in the “Discussion” section is again very useful and provides a bird’s eye view of how the gut microflora behave in various neurodevelopmental conditions. The authors discuss the three taxa that are differentially enriched or depleted across all neuronal pathologies discussed in this review. The rest of the section discusses the potential metabolic pathways and metabolites produced by these three taxa and how they feed into the BGM axis. Figure 2 does a really good job of representing the most promising microbial targets and how they might be involved in neuronal regulation via a feed back loop.
The authors conclude correctly that, even though these studies and others involving dietary interventions to manage epileptic seizures have shown a partial success, we need to adopt a cautious approach. There is a need for rigorous documentation and evaluation of neurological pathologies and the extent to which gut microbiome plays into these.
Overall, this is a well written review that provides a lot of useful information and perspectives for researchers interested in the correlation between neuronal pathologies and the BGM axis.
That said, the authors also need to include a few sentences to discuss the impact of antibiotic treatments on the gut microbiome. For example, many RTT patients are also on antibiotics on a regular basis due to recurrent lung infections. Such repetitive exposure to antibiotics can change the gut microbiome. In the two RTT studies highlighted in this review, it is possible that the difference in the microbial population between healthy individuals and RTT patients is due to antibiotic treatments and not due to the neuronal defects.
Author Response
REVIEWER 1
Q1- The authors also need to include a few sentences to discuss the impact of antibiotic treatments on the gut microbiome. For example, many RTT patients are also on antibiotics on a regular basis due to recurrent lung infections. Such repetitive exposure to antibiotics can change the gut microbiome. In the two RTT studies highlighted in this review, it is possible that the difference in the microbial population between healthy individuals and RTT patients is due to antibiotic treatments and not due to the neuronal defects.
A1- We thank the reviewer for the fruitful comment. Indeed, due to the importance of antibiotic exposure in manipulating gut microbiota, the antibiotic intake (up to three months before the enrollment) has been an exclusion criterion for both the considered studies on RTT syndrome. We have now better specified this aspect in the manuscript (lines 134-139), and added a reference on antibiotic use in RTT for recurrent lung infections (reference #45).
Reviewer 2 Report
Borghi and Vignoli with "RETT syndrome and Autism Spectrum Disorder share common changes in gut microbial community" presents a manuscript piece in the form of theoretical review for highlighting a common microbial denominator between RET syndrome and ASD.
My major concern resides in the form this paper was introduced to the readers. The title completely suggests a paper showing Borghi and Vignoli´s own data demonstrating this specific point (common changes in both disorders) or, at least, a complete systematic review of the literature with material and methods, exclusion and inclusion criteria and literature analysis, which finally may lead the authors to such conclusion.
The authors, instead, used a descriptive (and not systematical) theoretical review to analyze and compare two disorders with a final conclusion. Theoretical reviews may accidentally hide potential pitfalls or data in conflict with the author´s conclusion and should be considered beforehand depending on what we would like to analyze/compare.
Author Response
REVIEWER 2
Borghi and Vignoli with "RETT syndrome and Autism Spectrum Disorder share common changes in gut microbial community" presents a manuscript piece in the form of theoretical review for highlighting a common microbial denominator between RET syndrome and ASD.
My major concern resides in the form this paper was introduced to the readers. The title completely suggests a paper showing Borghi and Vignoli´s own data demonstrating this specific point (common changes in both disorders) or, at least, a complete systematic review of the literature with material and methods, exclusion and inclusion criteria and literature analysis, which finally may lead the authors to such conclusion.
The authors, instead, used a descriptive (and not systematical) theoretical review to analyze and compare two disorders with a final conclusion. Theoretical reviews may accidentally hide potential pitfalls or data in conflict with the author´s conclusion and should be considered beforehand depending on what we would like to analyze/compare.
A1- We agree with the reviewer that systematic reviews are characterized by methodological rigor and can benefit from guidelines such as PRISMA, reducing biased conclusions. However, a systematic review on Autism Spectum Disorder has been published very recently (Liu et al., Transl Psychiatry. 2019) and the median estimated validity is 3 to 5 years (Ferrari, Medical Writing 2015). Concerning Rett syndrome, the only two papers focusing on gut microbiota characterization (the third – Strati et al., BMC Gastroenterol. 2018- focusing only on Candida parapsilosis) do not allow performing a systematic review on the topic. The same is true for Down Syndrome, with only one study analyzing the gut microbiota in these patients. Thus, we feel that the main weakness ascribed to narrative review, i.e. the subjectivity in study selection, in this specific case is mitigated by the inclusion of all the available studies on the topic.
Not to mislead the readers, we specified in both title and abstract (line 12) the descriptive/narrative feature of the review. Moreover, we included the systematic review on ASD in the cited literature (lines 74 and 388-389).
Round 2
Reviewer 2 Report
The authors performed a number changes and language editing according to the referees´comments that to my mind significantly improved the quality of this manuscript. I think that the current piece could be accepted for publication.
I strongly suggest the authors to include as reference in their manuscript the recent systematic review of the literature by Argou-Cardozo & Zeidán-Chuliá, published in Medical Sciences (MDPI), which methodologically demostrates this link between ASD and Clostridium bacteria within this parragraph below:
<<Intriguingly, an altered composition of gut microbiome has been demonstrated in subjects with ASD [17-19], suggesting a possible link between dysbiosis (an aberrant shift in microbial ecology) and neurodevelopmental diseases>>
-----------------------------------------------------------------------------------------
Clostridium Bacteria and Autism Spectrum Conditions: A Systematic Review and Hypothetical Contribution of Environmental Glyphosate Levels.
Argou-Cardozo I, Zeidán-Chuliá F.
Med Sci (Basel). 2018 Apr 4;6(2). pii: E29. doi: 10.3390/medsci6020029. Review.
Author Response
Dear Reviewer,
We added the reference suggested.
Best regards,
Aglaia Vignoli